# Chromosome-specific barcode system with centromeric repeat in cultivated soybean and wild progenitor

Ahmet L Tek[1],*, Kiyotaka Nagaki[2],*, Hümeyra Yıldız Akkamış[1], Keisuke Tanaka[3], Hisato Kobayashi[3]

**Wild soybean *Glycine soja* is the progenitor of cultivated soybean *Glycine max*. Information on soybean functional centromeres is limited despite extensive genome analysis. These species are an ideal model for studying centromere dynamics for domestication and breeding. We performed a detailed chromatin immunoprecipitation analysis using centromere-specific histone H3 protein to delineate two distinct centromeric DNA sequences with unusual repeating units with monomer sizes of 90–92 bp (CentGm-1) and 413-bp (CentGm-4) shorter and longer than standard nucleosomes. These two unrelated DNA sequences with no sequence similarity are part of functional centromeres in both species. Our results provide a comparison of centromere properties between a cultivated and a wild species under the effect of the same kinetochore protein. Possible sequence homogenization specific to each chromosome could highlight the mechanism for evolutionary conservation of centromeric properties independent of domestication and breeding. Moreover, a unique barcode system to track each chromosome is developed using CentGm-4 units. Our results with a unifying centromere composition model using CentGm-1 and CentGm-4 superfamilies could have far-reaching implications for comparative and evolutionary genome research.**

## Introduction

*Glycine* is a genus in Fabaceae, and the genus has two subgenera, *Glycine* and *Soja*. The subgenus *Soja* includes two species, *Glycine max* (soybean) and *Glycine soja* (wild soybean). *G. max* is a commercially important crop that supplies vegetable protein and oil. On the contrary, *G. soja* is the closest wild species of *G. max*, and these two species have the same number of chromosomes, 2n = 2x = 40 (Hammatt et al, 1991). Both species are considered unique polyploid models of ancient paleopolyploidy origin and diverged after a genome duplication event estimated more than 15 million years ago (Schlueter et al, 2004). Genome sequencing of these species

revealed 0.3% small sequence differences including single nucleotide polymorphisms (SNPs) and insertions/deletions less than 35 bp and 3.5% large insertions/deletions greater than 100 bp between the species (Kim et al, 2010). A total of 17.3% and 0.2% of the deleted regions in *G. soja* were LTR retrotransposon–related and centromeric sequences, respectively.

Centromeres and kinetochores are essential complexes for accurately distributing chromatids to daughter cells during mitosis and meiosis (Amor et al, 2004). In these complexes, CENP-A/centromere-specific histone H3 (CENH3) is a key protein for assembling other kinetochore proteins and epigenetic determination of the kinetochore position (Perpelescu & Fukagawa, 2011). Because CENH3 directly binds to the centromeric DNA sequence as a component of core histones, centromeric DNA sequences have been identified in plant species by chromatin immunoprecipitation (ChIP) using anti-CENH3 antibodies (Zhong et al, 2002; Nagaki et al, 2003, 2004, 2009, 2011, 2012a, 2012b, 2015; Nagaki & Murata, 2005; Houben et al, 2007; Tek et al, 2011; Wang et al, 2011; Gong et al, 2012; Neumann et al, 2012; He et al, 2015; Marques et al, 2015; Ishii et al, 2020; Huang et al, 2021; Liu et al, 2021; Xue et al, 2022). In most cases, species-specific satellites and retrotransposons have been identified as centromeric sequences. In our previous soybean research, two satellite superfamilies (CentGm-1 and CentGm-4) and a retrotransposon-related sequence (GmCR) were isolated by ChIP cloning. However, comprehensive and quantitative analyses of centromeric DNA sequences were not included in the report (Tek et al, 2010) (Table 1).

There are several whole-genome assemblies available for *Glycine* species such as *G. max* (Schmutz et al, 2010), *G. soja* (Kim et al, 2010), *Glycine latifolia*, and *Glycine tomentella* (Liu et al, 2018). These assemblies do not show an accurate centromere structure that integrates functional centromeres. Therefore, the presence and boundaries of centromeres in *Glycine* species are still unclear. In this study, we characterized the *CENH3* gene of *G. soja* by sequencing and immunostaining with a previously reported anti-GmCENH3 (CENH3 from soybean) antibody. Because the antibody showed centromeric signals on *G. soja* chromosomes, a ChIP-Seq experiment was also conducted for a comprehensive analysis of DNA sequences that coexist with CENH3 in

[1]Department of Agricultural Genetic Engineering, Ayhan Şahenk Faculty of Agricultural Sciences and Technologies, Niğde Ömer Halisdemir University, Niğde, Türkiye [2]Institute of Plant Science and Resources, Okayama University, Kurashiki, Japan [3]NODAI Genome Research Center, Tokyo University of Agriculture, Setagaya, Japan

Correspondence: altek2@gmail.com; nagaki@rib.okayama-u.ac.jp
*Ahmet L Tek and Kiyotaka Nagaki contributed equally to this work

**Table 1.  Unifying model and nomenclature for DNA sequence composition of *Glycine* centromeres.**

| *Glycine* centromere composition (superfamily) | Name (subfamily) | Characteristics | References |
|---|---|---|---|
| CentGm-1 superfamily repeat (All chr) | SB92 | 92 bp length | Vahedian et al (1995) |
| | CentGm-1 (14 chr) | Nonoverlapping sets of chromosomes by FISH, 92 bp length, 80% similarity with CentGm-2 | Gill et al (2009) and Tek et al (2010) |
| | CentGm-2 (8 chr) | Nonoverlapping sets of chromosomes by FISH, 91 bp length, 80% similarity with CentGm-1 | |
| | CentGm-1a, CentGm-1b | The patterns of satellite repeats of the CentGm-1 superfamily identified four subfamilies with phylogenetic analysis | Kim et al (2021) |
| | CentGm-2a, CentGm-2b | | |
| | CentGm-91 | ChIP-Seq | Liu et al (2023) |
| | CentGm-92 | | |
| | CentGm-413 | | |
| CentGm-4 superfamily repeat (All chr) | Monomer | ~413 bp length | Tek et al (2010) |
| | Dimer | Specific genome organization that varies from chromosome to chromosome | This study |
| | Trimer | Specific genome organization that varies from chromosome to chromosome | This study |
| | Tetramer | Specific genome organization that varies from chromosome to chromosome | This study |
| | Pentamer | Specific genome organization that varies from chromosome to chromosome | This study |
| | CentGm-413 | ChIP-Seq | Liu et al (2023) |
| Centromeric retrotransposon (LTR) | GMCR | ChIP-cloning | Tek et al (2010) |
| | CL26 | ChIP-Seq | Liu et al (2023) |
| | CL33 | | |
| Tandem repeat (Chr1) | CentGm-273 | ChIP-Seq, different length and similarity from other centromeric satellites | Liu et al (2023) |
| | CentGm-444 | | |

the species. In addition, ChIP–qPCR confirmed the coexistence of the orthologous sequences and GmCENH3 in *G. max*. By comparing the functional centromere structure of a progenitor and its cultivated form, we hypothesized that centromere-specific DNA sequences are homogenized within a progenitor genome and whether such sequence homogenization is affected by domestication and breeding. Therefore, our study in *G. max* and *G. soja* compares the functional centromere structures between a cultivated species and its progenitor with a new nomenclature of a unifying centromere model including two distinct superfamily repetitive DNA sequences.

## Results

### Isolation and sequence analyses of CENH3 in *G. soja*

To determine the centromere-specific histone H3 homolog of *G. soja*, the amino acid sequence of *CENH3* cDNA (GmCENH3) from *G. max* (Tek et al, 2010) was used as a query in a TBLASTN search against the EST database. In addition, using conserved sequences (Table S1),

5′- and 3′-RACE–PCR experiments were performed to determine the expression of *G. soja* transcription. As a result, a putative full-length cDNA containing a 477-bp cDNA sequence encoding 159 amino acids (GenBank accession number: OP605950) was obtained. The deduced amino acid sequence showed the highest similarity (100%) with the GmCENH3 sequence (GenBank accession number: OP605951). The determined GsCENH3 amino acid sequence was aligned with the GmCENH3 sequence of cultivated soybean (Fig S1). The GsCENH3 protein sequence is highly conserved, both at the N-terminus and at the C-terminus. The comparison of CDS sequences between *GsCENH3* and *GmCENH3* revealed three nucleotide changes that did not cause any amino acid changes. According to the generated phylogenetic tree, CENH3s of *G. soja* and *G. max* were classified in the same branch. GsCENH3 showed the closest relationship with PvCENH3 from the legume species (Fig 1A).

### Centromeric DNA sequences in *G. soja* are coprecipitated with the GmCENH3 antibody

A previously raised GmCENH3 antibody was used to test whether the antibody recognizes the GsCENH3 protein at functional

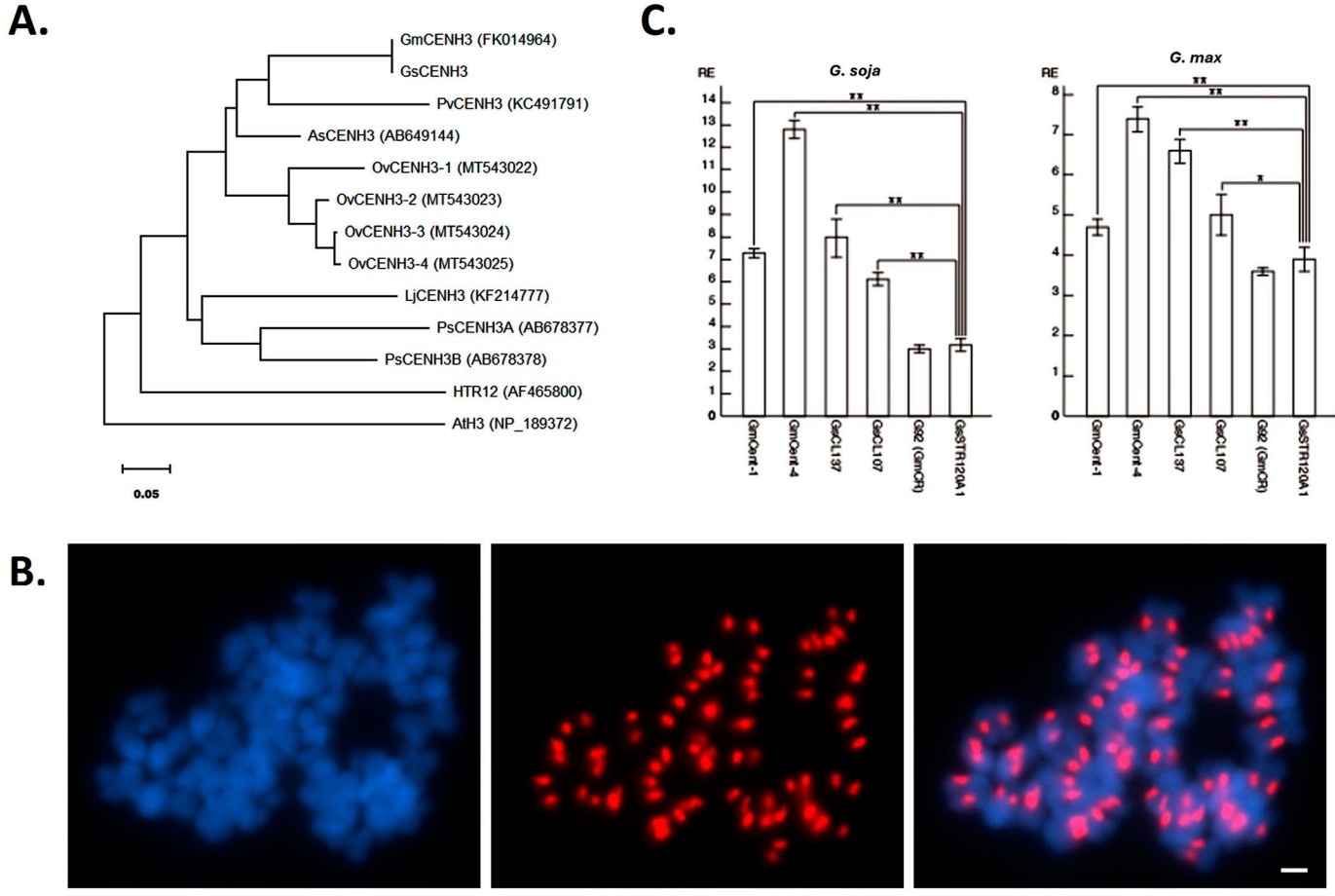

**Figure 1.  Phylogenetic relationship of the GsCENH3 protein with legume species, immunolocalization of the GsCENH3 protein, and analysis of centromeric DNA sequences by chromatin immunoprecipitation (ChIP).**
**(A)** Phylogenetic tree based on the amino acid sequences of legume CENH3. The phylogenetic tree was obtained using the neighbor-joining (NJ) model. *Arabidopsis thaliana* canonical histone H3 (AtH3) was used as an outgroup; HTR12 (*A. thaliana*), PsCENH3 (*Pisum sativum*), PvCENH3 (*Phaseolus vulgaris*), OvCENH3 (*Onobrychis viciifolia*), LjCENH3 (*Lotus japonicus*), AsCENH3 (*Astragalus sinicus*). **(B)** Immunostaining of *G. soja* metaphase chromosomes using an anti-GmCENH3 antibody. DAPI-stained chromosomes (blue), immunosignals of an anti-GmCENH3 antibody (red), and merged image of DAPI-stained chromosomes and immunosignals of an anti-GmCENH3 antibody. The scale bar is 10 $\mu$m. **(C)** ChIP–qPCR analysis of centromeric DNA sequences of *G. soja* and *G. max*. The columns and error bars represent the average relative enrichment (RE) and the SEs from four independent ChIP reactions, respectively. A repetitive sequence of soybean (STR120A.1, GenBank accession number: U26697) was used as a noncentromeric (negative) control in the anti-GmCENH3 ChIP. The statistical significance of differences between the negative controls and other sequences was determined using a $t$ test (\*\*$P < 0.01$ and \*$P < 0.05$).

centromeres in *G. soja*, because 20 of 20 residues are identical at the antigen polypeptide between GsCENH3 and GmCENH3 (Fig S1). The GmCENH3 antibody produced centromeric signals on all *G. soja* chromosomes (Fig 1B), indicating that the GmCENH3 antibody specifically recognizes the GsCENH3 protein in this species.

Subsequently, a ChIP-Seq experiment was conducted using the anti-GmCENH3 antibody and chromatin extracted from *G. soja* leaves to investigate DNA sequences that coexist with GsCENH3. DNA fragments from the input and the GmCENH3 fraction in the ChIP were sequenced using MiSeq with the paired-end 2 × 300 bp protocol. After the initial quality checks, 1,372,380 paired reads from the input and 983,560 paired reads from the GmCENH3 fraction were analyzed using the RepeatExplorer2 program. A total of 230 clusters containing at least 0.01% of the sequences used were generated by this analysis. Then, enrichment ratios (ERs) were calculated by the following formula: ER = GmCENH3 ChIP reads/input reads in each cluster, and the clusters were sorted by ER (Table S2). Of the 230 clusters formed, 12 genomic clusters had an ER greater than 2.0. The 12 clusters involved homologs of previously reported soybean centromeric tandem DNA sequences (four clusters each of CentGm-1 and CentGm-4), but a previously reported centromeric retrotransposon, GmCR, did not (Table S2). The percentages of the CentGm-1– and the CentGm-4–related clusters in the input reads were 3.6% and 0.6%, respectively. In addition, three gypsy-related clusters (GsCL87, GsCL107, and GsCL137) were involved in the 12 clusters, and these clusters were classified into two different groups (GsCL137 type and GsCL107 type involving GsCL87). The GsCL107 type showed 80% sequence similarity to a centromeric clone of *Lotus japonicus* (GenBank accession number: AF390569). The remaining cluster (GsCL224) did not show sequence similarity to any defined sequences in GenBank.

To confirm the ChIP-Seq results, ChIP–qPCR was conducted using four types of DNA sequences (CentGm-1, CentGm-4, GsCL107, and GsCL137) selected from the 12 clusters with an ER greater than 2.0, a GmCR (G92), and a homolog (GsSTR120A.1) of a noncentromeric repetitive sequence of soybean (STR120A.1, GenBank accession number: U26697) as a negative control (Fig 1C). The DNA sequences from all four types of DNA sequences from the 12 clusters were significantly enriched (7.3-fold for CentGm-1, 12.8-fold for CentGm-4, 8.0-fold for GsCL137, and 6.1-fold for GsCL107) compared with the negative control in the CENH3 fractions ($P < 0.01$ via a $t$ test, n = 4), whereas a previously reported centromeric retrotransposon, GmCR, was not enriched (3.0-fold, $P = 0.10$ via a $t$ test, n = 4), indicating that the ChIP–qPCR data showed the same tendency as the ChIP-Seq data.

In addition, a set of ChIP–qPCR was conducted using the same primers used for the qPCR and ChIP DNA from soybean (Fig 1C). In the soybean ChIP–qPCR data, homologs of the four significantly enriched *G. soja* sequences were also significantly enriched (4.7-fold for CentGm-1, 7.4-fold for CentGm-4, 6.6-fold for GsCL137, and 5.0-fold for GsCL107) compared with the negative control in the CENH3 fractions ($P < 0.01$ in CentGm-1, CentGm-4, and GsCL137 and $P = 0.02$ in GsCL107 via a $t$ test, n = 4), whereas a previously reported centromeric retrotransposon, GmCR, was not enriched (3.6-fold, $P = 0.23$ via a $t$ test, n = 4), indicating that these four sequences are centromeric in both species. Also, PCRs were performed in soybean and *G. soja* using primers specifically designed for CentGm-1 and CentGm-4 repeat sequences. Although the CentGm-1 repeat showed a smear band pattern because of its small monomer size, the CentGm-4 repeat formed a ladder band pattern (Fig S2).

## Two unrelated DNA sequences with different monomer sizes are localized on the centromeres of *G. soja* chromosomes

To confirm the centromeric localization of the DNA sequences immunoprecipitated with anti-GmCENH3 via the ChIP-Seq and ChIP–qPCR, these sequences were used as probes for fluorescence in situ hybridization (FISH) analysis (Fig 2A–D). Probes containing the CentGm-1 and CentGm-4 showed centromeric signals on chromosomes (Fig 2D). However, we could not obtain any FISH signals from GsCL137 or GsCL107 because of their low frequency (<0.2%) in the genome.

According to the FISH analysis performed on the *G. soja* metaphase chromosome set, the presence of all 40 chromosomes in the centromere region was determined as a result of biotin probe labeling representing the CentGm-1 family. Twelve chromosome sets showed major signals, whereas eight chromosome sets showed minor signals in *G. soja* centromeres. As a result of DIG probe labeling representing the CentGm-4 family, the presence of different signal intensities in the centromere region of all 40 chromosomes was detected. However, the major signals were detected on only seven chromosome sets (Fig 2C and D). Centromeric repeats were detected, especially in the heterochromatic regions of interphase nuclei stained with DAPI (Fig 2C). Both repeat sequences coexist in 13 chromosome sets. The presence and concentration of CentGm-1 and CentGm-4 repeat sequences varied from chromosome to chromosome. The number of strong signals obtained in both repeat sequences was the same as that obtained

in *G. max*. This similarity to *G. soja* confirms that *G. soja* is the ancestor and closest wild form of cultivated *G. max*. In addition, the results of immuno-FISH analysis revealed colocalization of the anti-GmCENH3 antibody with CentGm-1 in the interphase and metaphase chromosomes of *G. soja* (Fig 2A and B).

## Genome-wide bioinformatics analysis of two distinct centromeric DNA sequences in *G. max* and *G. soja* chromosomes

Genome-wide in silico analysis was performed to determine how conserved the CentGm-1 and CentGm-4 superfamily sequences were in *G. max* and *G. soja*. Centromeric repeat sequences of both plants were screened via whole-genome sequencing. In addition, sequences from the SRA database (*G. max*, SRR20982584; *G. soja*, SRR19807563) in the NCBI GenBank were supported by the TAREAN pipeline of RepeatExplorer2 (Novák et al, 2013, 2017), the presence of the CentGm-1 and CentGm-4 superfamily sequences in the whole genome (Fig S3). According to the results of RepeatExplorer, 50 consensus monomers were randomly selected from the CentGm-1 and CentGm-4 superfamilies. The consensus sequences were subjected to multiple nucleotide alignment and phylogenetic tree construction.

Phylogenetic analysis of the consensus monomers identified from the *G. max* and *G. soja* revealed two clades of subfamilies of the CentGm-1 superfamily. Similarly, two clade subfamilies were detected in the CentGm-4 repeat family (Fig 3). According to previous studies in soybeans, the CentGm-1 repeat superfamily was divided into 2 subfamilies based on phylogenetic tree results (Gill et al, 2009; Liu et al, 2018). Only one subfamily of the CentGm-1 repeat superfamily was found in the *G. latifolia*, *G. tomentella*, and *Glycine falcata* wild species (Liu et al, 2018; Zhuang et al, 2022). Surprisingly, in a recent bioinformatics-based study, the CentGm-1 centromeric repeat superfamily of the soybean cultivar Hwanggeum was divided into four subfamilies (Kim et al, 2021) (Table 1). However, no other study has supported this finding. Because two clade subfamilies were identified for the first time in the CentGm-4 repeat family, a detailed genome-wide analysis was performed here.

Genome-wide comprehensive analysis of centromeric satellite repeat sequences was performed in *G. soja* and *G. max*. To determine the specific distribution of each centromeric repeat on chromosomes, we used the whole-genome assemblies of the *G. soja* and *G. max* genomes from publicly available data (https://blast.ncbi.nlm.nih.gov) (Wang et al, 2021; Yi et al, 2022). The soybean Williams 82 chromosome system was ordered and constructed with genetic and physical maps based on the genetic markers (Jaffe et al, 2003; Schmutz et al, 2010). Genomic analysis was performed based on ChIP-Seq sequences, and 50 tandem monomers were selected from each chromosome to cover different points of the centromere. Monomers of repeat sequences selected specifically for chromosomes were subjected to multiple sequence alignment. The CentGm-1 superfamily, which consists of monomers with a maximum length of 93 bp, a minimum length of 88 bp, and averages of 91 and 92 bp, was found in both *G. max* and *G. soja*. On the contrary, the CentGm-4 superfamily, which consists of monomers with a maximum length of approximately 437 bp, a minimum length of 395 bp, and an average length of 413 bp, was

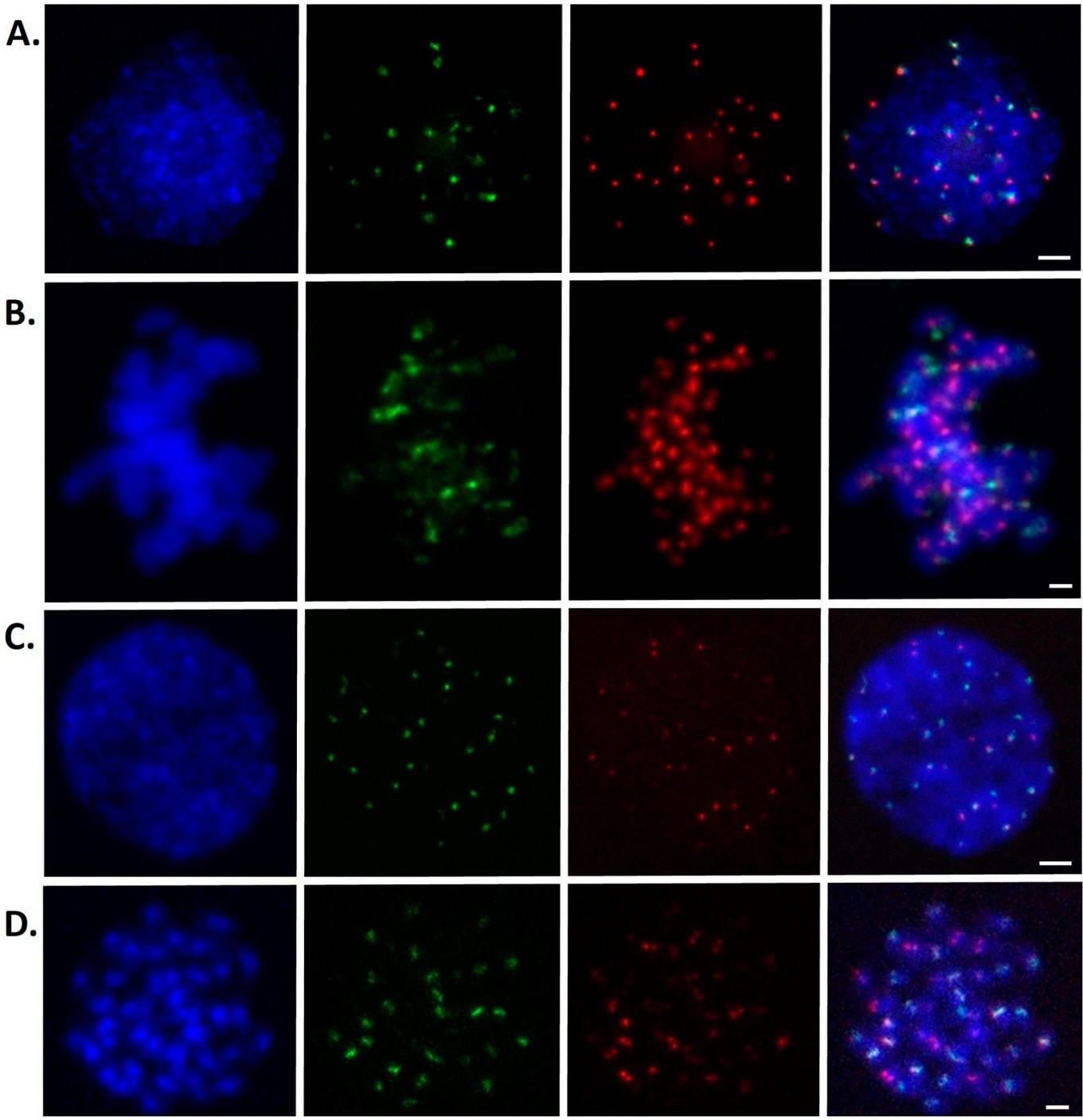

**Figure 2.   Fluorescence in situ hybridization (FISH) analysis of centromeric satellite repeat sequences, and colocalization of GsCENH3 antibody–CentGm-1 probes by immuno-FISH analysis on *G. soja* nuclei and metaphase chromosomes.**
Colocalization of the anti-GmCENH3 antibody with CentGm-1 in the interphase and metaphase chromosomes of *G. soja* is revealed. DAPI-stained interphase nuclei and metaphase chromosomes are shown in blue. **(A, B)** Centromeres detected on interphase (A) and metaphase (B) chromosomes with anti-GmCENH3 antibody (red signals). The same cells were hybridized with a biotin-labeled CentGm-1 probe (green signals) showing the colocalization of the immuno- and FISH signals on interphase and metaphase chromosomes. Localization of centromeric DNA sequences on *G. soja* metaphase chromosomes by FISH. **(C, D)** CentGm-1 and CentGm-4 probes are detected with biotin (green signals) and digoxigenin (red signals), respectively, on interphase nuclei (C) and metaphase chromosomes (D). The scale bar is 10 *μm*.

found in both *G. max* and *G. soja*. According to the multiple nucleotide alignment of monomer sequences from the CentGm-1 and CentGm-4 repeat superfamilies in *G. max*, approximately 71.4% and 70% pairwise similarity was observed, respectively, whereas 70.8% and 70.2% pairwise similarity was observed in *G. soja*.

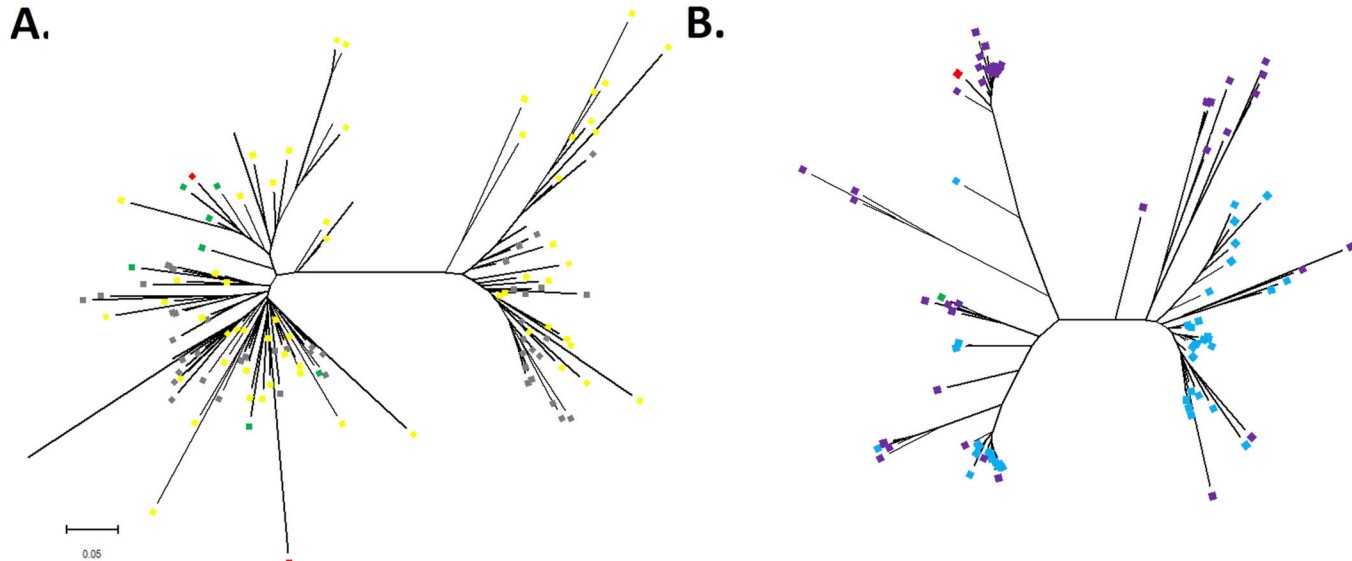

**Figure 3. Phylogenetic trees of centromeric satellite repeat monomers identified from *G. max* and *G. soja*.**
**(A)** Total of 50 consensus monomers from the *G. max* CentGm-1 family (yellow squares) and 50 consensus monomers from the *G. soja* CentGm-1 family (gray squares) were included in this analysis. **(B)** Total of 50 consensus monomers from the *G. max* CentGm-4 family (blue squares) and 50 consensus monomers from the *G. soja* CentGm-4 family (purple squares) were included in this analysis. **(A, B)** Monomers (green squares) from the plasmid clone used in the fluorescence in situ hybridization probe. Monomers (red squares) of CentGm-1 and CentGm-4 clones were obtained from Tek et al (2010).

According to the results of our extensive analysis, two subfamilies of the CentGm-1 centromeric repeat superfamily were detected at the chromosome level in both species. These subfamilies were named CentGm-1 and CentGm-2 based on conserved changes in monomer length and sequence motif, respectively. The CentGm-1 subfamily contains 92-bp monomers and is more abundant in the genome than the other centromeric repeat subfamily, that is, CentGm-2. On the contrary, the CentGm-2 subfamily consists of 91-bp monomers. Both different repeat families organize independently of each other in centromeric regions. CentGm-1 was detected in the centromeric region of a total of 16 chromosome sets. The CentGm-2 repeat is located in only nine chromosome sets. Both CentGm-1 and CentGm-2 repeats coexist in only five chromosome sets (Fig 4A and C). In cultivated and progenitor *Glycine* species, the CentGm-1 repeat superfamily showed the same organization at the chromosome level. This indicates that the CentGm-1 repeat superfamily is evolutionarily highly conserved. According to three different studies conducted in *G. max*, CentGm-1 and CentGm-2 repeats were detected in the centromeric region by the FISH technique. Gill et al (2009) detected CentGm-1 in 14 chromosome pairs, CentGm-2 in eight pairs, and both CentGm-1 and CentGm-2 repeat sequences in one chromosome pair together (Gill et al, 2009; Y. Liu et al, 2023). Findley et al (2010) detected CentGm-1 in 12 chromosome pairs, CentGm-2 in seven pairs, and both CentGm-1 and CentGm-2 repeat sequences in one chromosome pair together (Findley et al, 2010). The main reason for the difference in the number of centromeric loci of repeat sequences in *G. max* and *G. soja* could be that they may not be accurately detected by FISH because of the low percentage of repeat sequences in some chromosomes in *G. max*.

The other centromeric repeat sequence, the CentGm-4 superfamily, is present on all chromosomes, but its monomer organization is different from that of CentGm-1 in both species (Fig 4A and C). The CentGm-4 repeat superfamily is arranged in a higher order repeat (HOR) structure that varies from chromosome to chromosome (Fig 4B and D). However, the HOR structure of the CentGm-4 repeat superfamily is arranged differently on some chromosomes in *G. soja* and *G. max*. Units of the CentGm-4 repeat sequences are organized into independent tandem units of approximately 413/417 bp in *G. soja*. In only five sets of chromosomes, the CentGm-4 monomer was partially incorporated into the HOR structure, with lengths of 209, 252, and 317 bp. CentGm-4 is arranged into HORs by monomers in 11 different chromosome sets, dimers in seven different chromosome sets, trimers in four different chromosome sets, and tetramers in two chromosome sets. In addition, the CentGm-4 repeat superfamily showed chromosome-specific variation because of the presence of two different HOR characteristic structures in six different chromosome sets (Fig 4B and Table S3). Chromosomes were grouped according to the unit number of the characteristic HOR structure contained in the chromosomes, and subsequently, a chromosome-specific phylogenetic tree was constructed. Surprisingly, three different trees were formed from HORs grouped as monomers, dimers, and trimers. The tetramer is not included in the trees because it is found on only two chromosomes. Each chromosome on the tree was coded with a specific color, and the CentGm-4 HOR characteristic distribution, which allowed discrimination from chromosome to chromosome, was performed. According to the phylogenetic tree, chromosomes, especially those with dimer and trimer units, are characteristically and independently separated on the tree branches. Similarly, it is present in chromosomes with monomer units independently

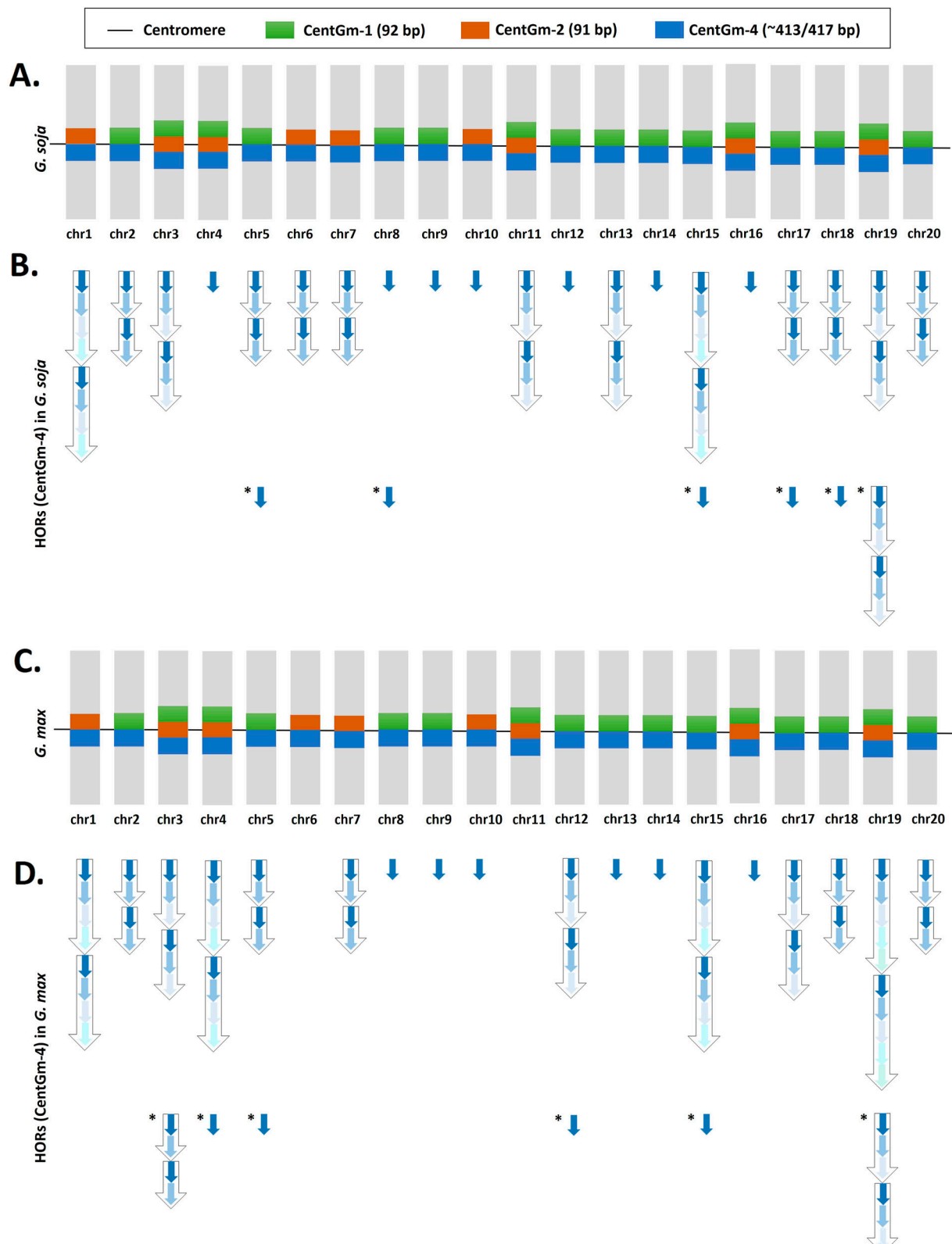

**Figure 4. Structure of centromeric DNA sequences and unique pattern of higher order repeat organization in *G. soja* and *G. max*.**
**(A)** Representative schematic organization of the CentGm-1 and CentGm-4 centromeric DNA sequence families in the genome of *G. soja* (A). CentGm-4 families are organized into HORs in *G. soja*. **(B, C)** Representative schematic organization of the CentGm-1 and CentGm-4 centromeric DNA sequence families in the genome of *G. max* (C). **(D)** CentGm-4 families are organized into HORs in *G. max* (D). The green, brown, and blue regions represent the CentGm-1 (92 bp), CentGm-2 (91 bp), and CentGm-4

branched, but it is present in some chromosomes that are closely related to other chromosomes scattered on the tree (Fig 5).

Units of the CentGm-4 repeat sequences are organized into independent tandem units of ~413/417 bp in *G. max*, similar to those in *G. soja*. In only four sets of chromosomes, the CentGm-4 monomer was partially incorporated into the HOR structure with lengths of 209, 252, and 317 bp. CentGm-4 is arranged into HORs by monomers in 10 different chromosome sets, dimers in six different chromosome sets, trimers in four different chromosome sets, tetramers in three chromosome sets, and pentamers in one chromosome set. In addition, the CentGm-4 repeat family showed variation because of the presence of two different HOR charac-teristic structures in six different chromosome sets (Fig 4D and Table S4). In conclusion, a detailed genome-wide sequence anal-ysis of CentGm-4 repeats provides evidence of chromosome-specific homogenization of CentGm-4 on nearly all chromosomes in *G. soja* and *G. max*.

According to our results based on genome-wide in silico anal-ysis, subfamilies of both repeat families were detected for the first time in both cultivated soybean and its progenitor. Both phylo-genetic analysis and multiple sequence alignment suggested a closer phylogenetic relationship of centromeric repeats between *G. soja* and *G. max*. The identification of two distinct satellite repeat families with no sequence homology to each other (CentGm-1 and CentGm-4) and their subfamilies at centromeres supports the hypothesis that soybean is an allopolyploid and has undergone a whole-genome duplication.

## Discussion

Centromeres are functionally conserved in eukaryotes because of the movement of chromosomes during cell division. This functional conservation contrasts with the highly variable DNA sequence composition leading to the postulation of centromere drive.

In the literature, centromeric DNA sequence comparisons for cultivated and wild species may provide a comparison of the evolutionary relatedness of species. For example, carrot (cultivated *Daucus carota*), which belongs to the Apiaceae family, has ~40 wild species. By targeting the CentDc centromeric repeat, which consists of a 39–40-bp monomer detected in carrot, the presence of the related repeat sequence in other *Daucus* species was screened with a FISH-based approach. The CentDc repeat, which is located in the primary constrictions of all chromosomes in carrot, is present in all chromosomes of some *Daucus* species, whereas it has been de-tected only in a certain number of chromosomes of some species. It is hypothesized that the CentDc repeat is common in the genus *Daucus* (Kadluczka & Grzebelus, 2021). With SbCENH3 ChIP-Seq analysis in sorghum species, the presence of the detected re-peat sequence was confirmed by genome-wide FISH analysis. The allotetraploid *Sorghum halepense* and one of its diploid parents, *Sorghum bicolor* centromeres, contain the abundant satellite

SorSat137 (CEN38), and various centromeric retrotransposons, specifically Ty3_gypsy-Athila and Ty1_copia-SIRE LTR. This high degree of similarity in centromeric composition indicates a close relationship between the two species and supports the hypothesis that *S. bicolor* may have played a role in the formation of *Sorghum halepense* (Kuo et al, 2021).

According to a previous study, the presence of the CentGm-1 repeat subfamily from *G. max* was also screened in wild species; nevertheless, its presence only in *G. soja* was supported by ex-periments with Southern blotting (Gill et al, 2009). However, the existence of the CentGm-1 repeat family in other *Glycine* species was also partly indicated by subsequent studies with limited evi-dence. The presence of centromeric repeats in different *Glycine* species has been determined by current bioinformatics analysis. CentGm-1 repeat superfamily-like sequences were found in species with different ploidy levels and genome groups, such as *G. latifolia*, *G. tomentella*, *G. falcata*, *Glycine syndetika*, *Glycine dolichocarpa*, *Glycine cyrtoloba*, and *Glycine stenophita*, whereas no evidence for the CentGm-4 repeat superfamily was presented. In addition, the centromeric retrotransposon family present in *G. max* has not been identified in perennial *Glycine* species (Liu et al, 2018; Zhuang et al, 2022). These findings indicate that the centromeric satellite DNA sequence in the genus *Glycine* is conserved and transferred be-tween species. In this study, two subfamilies of high-copy cen-tromeric satellite repeats, CentGm-1 and CentGm-4, were characterized in *G. soja* by integrating studies to date and cyto-logical strategies. The CentGm-4 repeat sequence, which was first detected in *G. max*, was also detected in *G. soja* for the first time among wild species. The centromeric CentGm-1 and CentGm-4 repeat sequences, which are present at different densities on all *G. soja* chromosomes, indicate that the genome of the wild ancestor was allopolyploid. The localization of the CentGm-1 and CentGm-4 repeat sequences on the same chromosomes indicates the pres-ence of these sequences in both ancestral parents. Sequencing of the *G. soja* genome revealed only a 0.31% difference from that of *G. max*. This result indicates that the whole genome, including the repeat sequences, has been transferred to the cultivated form of *G. max*, which is highly conserved (Kim et al, 2010).

Our data indicate that CentGm-1 and CentGm-4 are important parts of CENH3-associated chromatin in *G. max* and *G. soja*. It is possible that centromeric retrotransposon–related sequences are present at low abundance, which is below the detection limits of our study. Nevertheless, the presence of centromeric retrotransposon–related sequences has been indicated in previous studies (Tek et al, 2010; Neumann et al, 2011). Furthermore, the presence of other centromere-related DNA sequences cannot be ruled out.

Our study may bring soybean and its progenitor to the forefront of genomic and bioinformatics studies in the future because there are only a limited number of eukaryotic species with chromosome-specific markers associated with their centromeres at the whole-genome level. Using a single centromeric satellite repeat (CentGm-4) with a variable number of monomers creates an

repeat families, respectively. Arrows in different shades of blue show units of the HOR CentGm-4 family. * represents two variants that show two different genome compositions on the same chromosome.

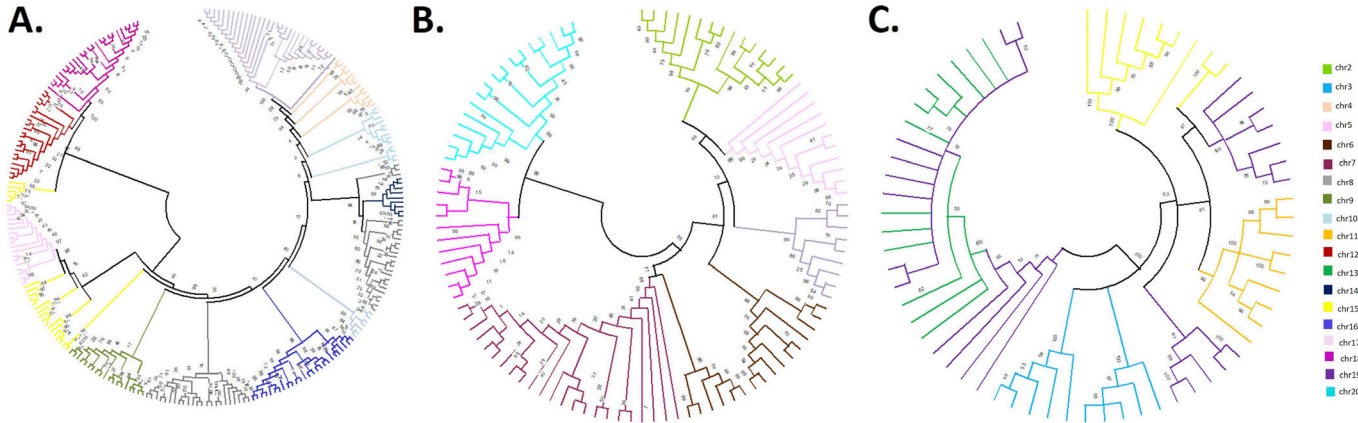

**Figure 5. Phylogenetic trees showing the chromosome-specific homogenization of the CentGm-4 repeat family in *G. soja*.**
**(A, B, C)** Monomer repeat units, (B) dimer repeat units, and (C) trimer repeat units, on chromosomes. Chromosome numbers are shown with their corresponding colors.

efficient and useful barcoding system for each centromere, thereby making it accessible to the whole genome at the individual chromosome level (Fig 4). Our unique chromosome barcode system to track individual chromosomes in both species is provided by CentGm-4 repeat units. Chromosome barcoding is used with centromeric satellite DNA repeats in cultivated and progenitor *Glycine* species. The centromeric repeat arrays in the same chromosome could potentially be highly variable among different individuals, genotypes, or cultivars. This may limit the potential application of using centromeric repeat HOR as a chromosome-specific barcode system. In addition, centromere regions containing tandem repetitive DNA sequences are inherently difficult to assemble even with advanced approaches, that is, next-generation sequencing. The low number of the CentGm-4 HOR units and shorter total length of CentGm-4 HOR per chromosome (mostly <2,000 bp) bring a substantial drawback. Nevertheless, our barcoding system requires proof-of-concept experimental refinement with specific combinations of CentGm-4 repeats for individual centromeres and probes for genome-scale hybridization detection across species. This process may be tested with current tools such as the design of genome-scale oligonucleotide in situ hybridization probes (Beliveau et al, 2018).

Our study contributes to a better understanding of the dynamic evolution of centromeric satellites present in other *Glycine* species. In addition, our study raises new questions regarding the possible ancestral relationships of soybean and its progenitor with other related wild *Glycine* species. The evolutionary origin of soybean can be further elucidated if the barcode system is applied to distantly related species in the future.

# Materials and Methods

### Plant material

Seeds of *G. soja* (2*n* = 2× = 40, accession numbers: JP110755 and CPI96904) were obtained from the National Institute of Agrobiological Sciences (NIAS) and the Australian Grains Genebank GRIN-Global Database System. Seeds of *G. max* cv. Sapporo Midori and cv. Adasoy (2*n* = 2× = 40) were obtained from commercial sources (LIC).

### Identification of a *G. soja* expressed sequence tag (EST) encoding CENH3

Nucleotide BLAST (BLASTN) analysis was performed with CDS against the whole-genome shotgun contigs (whole-genome sequencing) of *G. soja* (Phytozome; https://phytozome.jgi.doe.gov/pz/portal.html). Also, an EST sequence encoding *G. soja* CENH3 (GsCENH3) was identified from the gene indices using the TBLASTN program (https://blast.ncbi.nlm.nih.gov/Blast.cgi). The amino acid sequence of GmCENH3 (GenBank accession number: FK014964, Tek et al, 2010) was used as a query in both index searches.

### RNA isolation and PCR

Total RNA was extracted from 100 mg of young *G. soja* seedlings with an RNeasy Plant Mini kit (QIAGEN) according to the manufacturer's instructions. To determine the full-length cDNA sequence of the *G. soja CENH3* gene, rapid amplification of cDNA ends (RACE) was conducted using a SMARTer RACE 5′/3′ kit (Takara). For 3′-RACE, the primer GsCENH3-3′-RACE was designed from an EST encoding a putative *G. soja* CENH3 sequence determined via a BLAST search. Another primer, GsCENH3-5′-RACE, was designed from the sequences determined by 3′-RACE and was used to determine the 5′-end.

### Cloning of *GsCENH3* cDNA, sequencing, and sequence analyses

First-strand cDNA synthesis and rapid amplification of cDNA ends via 3′- and 5′-(RACE) PCRs were performed using the SMARTer RACE cDNA Amplification kit (Clontech) according to the manufacturer's instructions. The amplified cDNA product was cloned into a pGEM-T Easy vector (Promega) and sequenced from both ends using a BigDye Terminator v1.1 cycle sequencing kit and an ABI PRISM 3130xl genetic analyzer (Applied Biosystems). Bioinformatics analyses

were performed using Geneious Pro bioinformatics software (Biomatters). The GsCENH3 amino acid sequence was aligned with different plant species and canonical histone H3 using the Clustal X software program, and phylogenetic relationships were analyzed by the neighbor-joining method (Thompson et al, 1997).

## Immunostaining

Immunostaining was carried out using an anti-GmCENH3 antibody as previously described (Tek et al, 2010). According to protocol, root tips of *G. soja* were fixed in PHEMES buffer (50 mM PIPES, pH 6.9, 5 mM $MgSO_4$, and 5 mM EGTA) containing 3% (wt/vol) PFA and 0.2% (vol/vol) Triton X-100 for 20 min at 4°C. The fixed tips were digested with a mixture of 1% (wt/vol) Cellulase Onozuka RS (Yakult Pharmaceutical Industry) and 0.5% (wt/vol) Pectolyase Y-23 (Seishin Pharmaceuticals) in PHEMES buffer at 37°C. The digested meristematic tissue was squashed on poly-L-lysine–coated slides (Matsunami). Slides were incubated with a 1:100 dilution of the anti-GmCENH3 antibody in the TNB buffer at 4°C overnight. The anti-GmCENH3 antibody was incubated with Alexa Fluor 555–labeled anti-rabbit antibodies (Molecular Probes) at 37°C. Chromosomes were stained with DAPI. Signals on the chromosomes were captured using a fluorescence microscope equipped with a chilled charge-coupled device (CCD) camera, AxioCam HR (Carl Zeiss).

## ChIP

ChIP was performed as previously described (Nagaki et al, 2012a) with minor modifications using the anti-GmCENH3 antibody. Chromatin was isolated from the leaves of 1-mo-old *G. soja* and *G. max* plants by digestion with micrococcal nuclease (Sigma-Aldrich). After overnight incubation of the chromatin with the antibody at 4°C, the antibody was captured using Dynabeads Protein A (Invitrogen). Normal rabbit serum was used in replacement of the antibody for mock experiments. DNA was subsequently purified from the chromatin with the captured antibody by phenol/chloroform extraction followed by ethanol precipitation.

## ChIP-Seq and RepeatExplorer analysis

ChIP-Seq analysis was conducted using precipitated DNA from the input and the GmCENH3 fractions in the ChIP extracts. Libraries were constructed using the NEBNext ChIP-Seq Library Prep Reagent Set for Illumina (New England Biolabs). The libraries were processed by MiSeq (Illumina) with the paired-end 2 × 300 bp protocol. The raw base-call data were converted to sequence data in a fastq format, the reads derived from each sample were identified by index sequence, and adapter trimming was performed using MiSeq reporter 2.3.32. The DNA sequence data were analyzed by the similarity-based clustering program RepeatExplorer2 (http://repeatexplorer.org) with default parameters (Novák et al, 2010).

## qPCR

qPCR was conducted using SYBR Premix Ex Taq II (Tli RNaseH Plus; Takara) with a StepOne instrument (Applied Biosystems). The primers were designed based on the sequences in the clusters of the RepeatExplorer analysis. The specific primer pairs used for qPCR are provided in Table S1. The precipitated DNA in the ChIP experiment was used as a template, and the mock was used as a negative control. Relative enrichment (RE) was calculated by the following formula: RE = amount of the sequence in the antibody fraction/amount of the sequence in the mock. The qPCR results were assessed by a *t* test.

## FISH

FISH was conducted on *G. soja* nuclei and metaphase chromosomes as previously described (Tek et al, 2011). Approximately 1-cm-long *G. soja* root tips were collected and pretreated. The root tips were fixed in methanol: glacial acetic acid (3:1) overnight. Fixed root tips were digested in 0.5% (wt/vol) Cellulase Onozuka RS and 0.5% Pectolyase Y23 at 37°C for 1.5 h. Slide preparations from the digested root tips were made via the flame-drying method (Tek et al, 2011). Probe DNAs were labeled using biotin-dUTP with Biotin-Nick Translation Mix (Roche) and using digoxigenin-dUTP with DIG-Nick Translation Mix (Roche). Hybridized probes were then detected using streptavidin-conjugated Alexa Fluor 488 (Invitrogen) and rhodamine-conjugated anti-digoxigenin (Roche) antibodies. Chromosomes were stained with DAPI. Fluorescence signals on the chromosomes were captured using a fluorescence microscope at 63× magnification (Axio Imager.A2; Carl Zeiss). Images were captured with a monochromatic charge-coupled device camera (Axiocam 702; Carl Zeiss) operated with multichannel ZEN Pro imaging software.

## Bioinformatics analysis of centromeric repeats

To analyze chromosome-specific homogenization and determine the distribution of CentGm-1 and CentGm-4 centromeric repeat superfamilies in the genome, BLAST searches were conducted against 50× and 77× PacBio sequence coverage in *G. soja* cultivar F and *G. max* cultivar PI594527, respectively. Publicly available data were used to determine the distribution of centromeric repeats (https://blast.ncbi.nlm.nih.gov) (Wang et al, 2021; Yi et al, 2022). The full-length sequence of each chromosome was downloaded in a FASTA format. Using the DNA sequences obtained from the ChIP-Seq results, 50 different tandem monomers were identified from each chromosome. All monomers specific to chromosomes were processed for multiple sequence alignments using Clustal Omega in Geneious Pro v.4 (Biomatters) with default parameters. Phylogenetic analyses were performed by MEGA 11 with the neighbor-joining algorithm and the Tamura–Nei model. Bootstrap values were calculated from at least 1,000 replications (Thompson et al, 1997).

# Data Availability

All data were deposited in a publicly available repository NCBI with the following accession numbers: GsCenH3 OP605950, GmCenH3 OP605951, CentGm-1 OP605952, and CentGm-4 OP605953.

# Supplementary Information

# Acknowledgements

We thank the laboratory members for their excellent technical help during this research. This research was supported by the Cooperative Research Grant of the Genome Research for BioResource, NODAI Genome Research Center, Tokyo University of Agriculture, the Fellowship Programs of the Japan Society for the Promotion of Science (JSPS), Okayama University Research Funds, and the Scientific and Technological Research Council of Turkey (TÜBİTAK) Project ID 121O014.

## Author Contributions

AL Tek: conceptualization, resources, data curation, formal analysis, supervision, funding acquisition, validation, investigation, methodology, project administration, and writing—original draft, review, and editing.
K Nagaki: conceptualization, resources, data curation, formal analysis, funding acquisition, validation, investigation, visualization, methodology, project administration, and writing—original draft, review, and editing.
H Yıldız Akkamış: data curation, validation, investigation, visualization, methodology, and writing—review and editing.
K Tanaka: formal analysis, investigation, and methodology.
H Kobayashi: formal analysis, investigation, and methodology.

## Conflict of Interest Statement

The authors declare that they have no conflict of interest.

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
