## [Reviewer comments · Life Science Alliance]

Life Science Alliance

Chromosome-specific barcode system with centromeric repeat in cultivated soybean and wild progenitor

Ahmet Tek, Kiyotaka Nagaki, Hümeyra Yıldız Akkarnış, Keisuke Tanaka, and Hisato Kobayashi

DOI: 10.26508/lsa.202402802

Corresponding author(s): Ahmet Tek, Niğde Ömer Halisdemir Üniversitesi and Kiyotaka Nagaki, Okayama University

Review Timeline:	Submission Date:	2024-04-30
	Editorial Decision:	2024-06-04
	Revision Received:	2024-09-03
	Editorial Decision:	2024-09-17
	Revision Received:	2024-09-21
	Accepted:	2024-09-24

Transaction Report:

June 4, 2024

Re: Life Science Alliance manuscript #LSA-2024-02802-T

Prof. Ahmet L. Tek
NiÅŸde University
Department of Agricultural Genetic Engineering, NiÅŸde Ã–mer Halisdemir University
NiÅŸde Main Campus
Niğde 51240
Turkey

Dear Dr. Tek,

Thank you for submitting your manuscript entitled "Identification and comparison of functional centromere structure between cultivated and progenitor Glycine species: a chromosome-specific barcode system with a centromeric satellite DNA repeat" to Life Science Alliance. The manuscript was assessed by expert reviewers, whose comments are appended to this letter. We invite you to submit a revised manuscript addressing the Reviewer comments.

Thank you for this interesting contribution to Life Science Alliance. We are looking forward to receiving your revised manuscript.

Sincerely,

B. MANUSCRIPT ORGANIZATION AND FORMATTING:

Reviewer #1 (Comments to the Authors (Required)):

In this manuscript, Tek et al. identified two distinct centromeric DNA sequences with a monomer size of 90-92 bp (CentGm-1) and a monomer size of 413-bp (CentGm-4) by performed a detailed chromatin immunoprecipitation analysis using centromere-specific histone H3 protein. These two unrelated DNA sequences with no sequence similarity are part of functional centromeric chromatin in both species. They provided a comparison of centromere properties of two closely related species, a cultivated and a wild species under the effect of the same kinetochore protein. They also developed a unique chromosome barcode system to track individual chromosomes in both species is provided by using CenGm-4 repeat units. These results with a unifying centromere composition model provide well knowledge for genomic study in soybean species. The manuscript is well organized and written.

Specific comment:

The chromosome number system has not been well described in the manuscript.

Reviewer #2 (Comments to the Authors (Required)):

In this manuscript, the authors aimed to identify the centromeric repeat composition in the wild *Glycine soja*, and further compared the centromeric repeat composition between the cultivated *G. max* and the wild *G. soja*. Taking advantage of the identical CENH3 protein sequence in the two species, authors applied the available *G. max* anti-CENH3 antibody in *G. soja* for CENH3-ChIP sequencing and CENH3-ChIP qPCR. The identified centromeric repeat candidates were further confirmed cytologically by immuno-FISH. The methods and strategies are accurate and straight forward. According to the sequence similarity, the authors found the subfamilies of the centromeric repeats and analyzed the HOR structure of CentGm-4 in each chromosome in both species, suggesting using the chromosome-specific HOR of CentGm-4 as a barcode system for *Glycine* chromosome identification.

Comments and suggestions:

(1) The finding regarding the subfamilies of CentGm-1 CentGm-4 was recently reported in several *Glycine* species and cultivars in Liu et al., 2023. Additionally, comparing the HOR structure of CentGm-4 analyzed in a single individual of both *G. max* and *G. soya* was not representative in a species-level.

(2) I have a special concern about the application of the HOR variation of the centromeric CentGm4 repeats among different chromosomes as a barcode system for chromosome identification, especially by oligo painting as the authors proposed in the end of Discussion.

First, the centromeric repeat arrays in the same chromosome can be highly variable among different individuals of the same species. This will likely hamper the potential application of using centromeric repeat HOR as a chromosome-specific barcode system. Second, the number of repeat units is low and the total length of the CentGm-4 HOR is not long, mostly < 2000 bp among chromosomes. Therefore, the number of oligo probes might not be sufficient to generate chromosome-specific labelling pattern or detectable signals. Thus, using the variable HORs of the centromeric repeat arrays as a barcode system should be reconsidered or removed in the manuscript.

(3) The order of figures and tables in the main text should be mentioned and followed the alphabetic and numerical order. For example, the Fig. 1b show in the main text earlier than Fig. 1a, also Fig. 4a and 4b, and some Supplementary Tables and Figures were not mentioned in the main text, such as Suppl. Figure 2 and Suppl. Table 3...etc.

(4) In Figure 2, it will be better to show Fig. 2 c-d first, and Fig. 2 a-b, and add one more panel in Fig. 2 c-d to show the merge of CENH3 and CentGm-1 signals.

(5) The ides to unify the nomenclature of *Glycine* centromeric repeats is good, but many centromeric repeats reported in the recently published paper Liu et al., 2023 were not included in the Suppl. Table 5.

(6) English editing is required to concise and improve the quality of the manuscript.

Reviewer #1 (Comments to the Authors (Required)):

In this manuscript, Tek et al. identified two distinct centromeric DNA sequences with a monomer size of 90-92 bp (CentGm-1) and a monomer size of 413-bp (CentGm-4) by performed a detailed chromatin immunoprecipitation analysis using centromere-specific histone H3 protein. These two unrelated DNA sequences with no sequence similarity are part of functional centromeric chromatin in both species. They provided a comparison of centromere properties of two closely related species, a cultivated and a wild species under the effect of the same kinetochore protein. They also developed a unique chromosome barcode system to track individual chromosomes in both species is provided by using CenGm-4 repeat units. These results with a unifying centromere composition model provide well knowledge for genomic study in soybean species. The manuscript is well organized and written.

We are thankful to the reviewer for the positive comments and praise.

Specific comment:

The chromosome number system has not been well described in the manuscript.

The soybean genome assembly was constructed with genetic and physical maps based on the genetic markers. Twenty chromosome scaffolds of soybean Williams 82 were produced and ordered in position (Schmutz et al., 2010; Jaffe et al., 2003). Wang et al., (2021) and Yi et al., (2022) assembled the soybean and *Glycine soja* whole genome sequences, respectively, using the Williams 82 reference genome. The assembly sequences used in this manuscript were analyzed following the chromosome number system recommended in these articles. We described the chromosome number system with citations as suggested.

In the manuscript, how the chromosome number system is constructed and added under the following headings with the references. The revisions are marked with blue text.

- In the results “Genome-wide bioinformatic analysis of two distinct centromeric DNA sequences in *G. max* and *G. soja* chromosomes”.
- In material and methods “Bioinformatic analysis of centromeric repeats”.

Reviewer #2 (Comments to the Authors (Required)):

In this manuscript, the authors aimed to identify the centromeric repeat composition in the wild *Glycine soja*, and further compared the centromeric repeat composition between the cultivated *G. max* and the wild *G. soja*. Taking advantage of the identical CENH3 protein sequence in the two species, authors applied the available *G. max* anti-CENH3 antibody in *G. soja* for CENH3-ChIP sequencing and CENH3-ChIP qPCR. The identified centromeric repeat candidates were further confirmed cytologically by immuno-FISH. The methods and strategies are accurate and straight forward. According to the sequence similarity, the authors found the subfamilies of the centromeric repeats and analyzed the HOR structure of CentGm-4 in each chromosome in both species, suggesting using the chromosome-specific HOR of CentGm-4 as a barcode system for *Glycine* chromosome identification.

We thank the reviewer for the positive comments and praise.

Comments and suggestions:

(1) The finding regarding the subfamilies of CentGm-1 CentGm-4 was recently reported in several *Glycine* species and cultivars in Liu et al., 2023. Additionally, comparing the HOR structure of CentGm-4 analyzed in a single individual of both *G. max* and *G. soya* was not representative in a species-level.

We are grateful to the reviewer for this expert remark. We included the recent work from Liu et al., 2023. Also, the findings from this work are added to the Table 1 with proper reference to support our results and models. Comparing the HOR structure of CentGm-4 based on a single individual in both species might be a valid argument for not being representative in species level. However, current publicly available resources do not allow us to test additional

individuals/genotypes/cultivars since aforementioned data is not accessible in public domains from Liu et al. 2023. Therefore, we believe that testing additional individuals requires extensive financial and personal resources beyond the limitation of our study.

(2) I have a special concern about the application of the HOR variation of the centromeric CentGm4 repeats among different chromosomes as a barcode system for chromosome identification, especially by oligo painting as the authors proposed in the end of Discussion. First, the centromeric repeat arrays in the same chromosome can be highly variable among different individuals of the same species. This will likely hamper the potential application of using centromeric repeat HOR as a chromosome-specific barcode system. Second, the number of repeat units is low and the total length of the CentGm-4 HOR is not long, mostly < 2000 bp among chromosomes. Therefore, the number of oligo probes might not be sufficient to generate chromosome-specific labelling pattern or detectable signals. Thus, using the variable HORs of the centromeric repeat arrays as a barcode system should be reconsidered or removed in the manuscript.

This special concern has merits and many thanks for the expert remark. In chromosome-specific labeling, the copy number of the repeat units is quite effective rather than the total length of the HOR units. When examined in terms of copy number of CentGm-4 repeat, the copy number in most chromosomes is at least 20, therefore the total size covered by the repetitive DNA sequences is >1-2 kb. In special circumstances and under highly skilled labs, this length can be detected with chromosome-specific designed oligonucleotides and high-resolution techniques such as fiber-FISH. The fiber-FISH technique allows high-resolution mapping of chromatin fibers or DNA such as the physical ordering of DNA probes, assessment of gaps and overlaps in contigs, copy number variants, and small single or low copy DNA sequences in plants (Shakoori, 2017). Also, oligo-FISH mapping may permit the identification of low target sites like 1 kb target sites in plant species with low-copy oligo probes. Therefore, the HOR structure length found in the CentGm-4 repeat structure, which is >1 kb in soybean, can be detected using oligonucleotide probes.

Also, we added these critical points in the revised discussion section step-by-step pointing out the potential drawbacks and limitations of our study. Still, we believe that our complementary approaches support our main conclusions. These sections are highlighted in blue.

(3) The order of figures and tables in the main text should be mentioned and followed the alphabetic and numerical order. For example, the Fig. 1b show in the main text earlier than Fig. 1a, also Fig. 4a and 4b, and some Supplementary Tables and Figures were not mentioned in the main text, such as Suppl. Figure 2 and Suppl. Table 3...etc.

As advised by the reviewer, each figure and table reference has been checked in the main text. In the main text, all figures, tables and supplementary information are mentioned in alphabetical and numerical order. The Suppl. Fig. 2 was identified as missing, as recommended, and is included and updated in the main text. In addition, Figure 1 was reorganized and pictures 1A and 1B were replaced and designed to fit the chronological flow.

(4) In Figure 2, it will be better to show Fig. 2 c-d first, and Fig. 2 a-b, and add one more panel in Fig. 2 c-d to show the merge of CENH3 and CentGm-1 signals.

Thanks to the reviewer for his recommendation. As suggested, Figure 2 has been reorganized with the first 2 c-d images followed by 2 a-b images. Additionally, the merged image of the CENH3 antibody and CentGm-1 DNA sequence signals is updated as the fourth panel based on the recommendation.

(5) The ides to unify the nomenclature of Glycine centromeric repeats is good, but many centromeric repeats reported in the recently published paper Liu et al., 2023 were not included in the Suppl. Table 5.

Thanks to the reviewer for the remark. Suppl. Table 5 has been included as Table 1 in the main text. In addition, in the Liu et al., 2023 published manuscript, a total of 5 satellite repeats and 2 retrotransposons were identified. It was noticed that the repeat sequences CentGm91, CentGm92

and CentGm413 were not included in both tables. These repeats are updated where necessary in the tables as suggested.

(6) English editing is required to concise and improve the quality of the manuscript.

Thank the reviewer for the suggestions. We edited for shorter sentences, correct punctuation, singular/plural nouns, subject-verb agreement, proper flow of the story in the research. We also added proper citations where necessary. We hope that the revised version is satisfactory.

September 17, 2024

RE: Life Science Alliance Manuscript #LSA-2024-02802-TR

Prof. Ahmet L. Tek
Niğde University
Department of Agricultural Genetic Engineering, Niğde Omer Halisdemir University
Niğde Main Campus
Niğde 51240
Turkey

Dear Dr. Tek,

Thank you for submitting your revised manuscript entitled "A chromosome-specific barcode system with centromeric repeat in cultivated and progenitor soybean". We would be happy to publish your paper in Life Science Alliance pending final revisions necessary to meet our formatting guidelines.

- please address the Reviewer's remaining comments
- please be sure that the authorship listing and order is correct
- please add ORCID ID for secondary corresponding author-they should have received instructions on how to do so
- please use the [10 author names, et al.] format in your references (i.e. limit the author names to the first 10)
- please add a separate figure legend section to your main manuscript text
- we encourage you to introduce the figure panels in alphabetical order for figure 4

Figure Check:

- please add scale bars to Figure 1B

LSA now encourages authors to provide a 30-60 second video where the study is briefly explained. We will use these videos on social media to promote the published paper and the presenting author (for examples, see <https://docs.google.com/document/d/1-UWCfbE4pGcDdcgzcmiuJI2XMBJnxKYeqRvLLrLS08s/edit?usp=sharing>). Corresponding or first-authors are welcome to submit the video. Please submit only one video per manuscript. The video can be emailed to contact@life-science-alliance.org

A. FINAL FILES:

B. MANUSCRIPT ORGANIZATION AND FORMATTING:

Sincerely,

Reviewer #2 (Comments to the Authors (Required)):

1. I suggest removing "germplasm" from the keywords, since it is not the focus of this study, and add "centromeric repeat" and "Glycine"
2. Subgenus name should be italic.
3. Figure 1 legend, PvCENH3. Scale bar was mentioned in the legend, but not on the image.
4. Figure 2, please confirm if the scale bar can be applied in all four panels. Because according to this scale bar, the diameter of interphase nuclei can be almost 100 μm which is unexpected in the small-genome Glycine species.
5. Please confirm in Table 1, if the CentGm413 should belong to the CentGm-4 superfamily repeat.
6. I suggest a small modification of the title "A chromosome-specific barcode system using centromeric repeats in cultivated soybean and its wild progenitor".

Reviewer #2 (Comments to the Authors (Required)):

1. I suggest removing "germplasm" from the keywords, since it is not the focus of this study, and add "centromeric repeat" and "Glycine"

We are thankful to the reviewer. Based on the reviewer's suggestion, the keyword "germplasm" was removed and the keyword only "Glycine" was included in the manuscript since the max number of keywords is six. The title contains "centromeric repeat".

2. Subgenus name should be italic.

We are thankful to the reviewer. Throughout the manuscript, the subgenera *Soja* and *Glycine* of the genus *Glycine* are arranged in italics and marked in blue.

3. Figure 1 legend, PvCENH3. Scale bar was mentioned in the legend, but not on the image.

The scale in Figure 1 was overlooked. We thank the reviewer for his/her attention. The scale has been added to Figure 1. PvCENH3 in Figure 1 legend is corrected.

4. Figure 2, please confirm if the scale bar can be applied in all four panels. Because according to this scale bar, the diameter of interphase nuclei can be almost 100 μm which is unexpected in the small-genome *Glycine* species.

The scale bar in Figure 2d represents the panels in the entire Figure 2. Metaphase chromosomes and interphase nuclei images were taken from a fluorescence microscope at 63 \times magnification at 10 μm scale and organized. We added the scale bars to each panel. We checked and confirmed the details.

5. Please confirm in Table 1, if the CentGm413 should belong to the CentGm-4 superfamily repeat.

The CentGm413 belongs to the CentGm-4 superfamily repeat. Table 1 has been edited as necessary.

6. I suggest a small modification of the title "A chromosome-specific barcode system using centromeric repeats in cultivated soybean and its wild progenitor".

We are grateful to the reviewer for suggestions. However, there is a title limit for the journal with a 100 character limit (including spaces). We are not allowed to exceed the character limit in the submission system. We updated the title as follows: **Chromosome-specific barcode system with centromeric repeat in cultivated soybean and wild progenitor.**

September 24, 2024

RE: Life Science Alliance Manuscript #LSA-2024-02802-TRR

Prof. Ahmet L. Tek
Niğde Ömer Halisdemir Üniversitesi
Department of Agricultural Genetic Engineering
Nigde Omer Halisdemir University
Nigde 51240
Turkey

Dear Dr. Tek,

Thank you for submitting your Resource entitled "Chromosome-specific barcode system with centromeric repeat in cultivated soybean and wild progenitor". It is a pleasure to let you know that your manuscript is now accepted for publication in Life Science Alliance. Congratulations on this interesting work.

DISTRIBUTION OF MATERIALS:

Again, congratulations on a very nice paper. I hope you found the review process to be constructive and are pleased with how the manuscript was handled editorially. We look forward to future exciting submissions from your lab.

Sincerely,
